# Dendrimers-Based Drug Delivery System: A Novel Approach in Addressing Parkinson's Disease

Michaella B. Ordonio [1], Randa Mohammed Zaki [2,3] and Amal Ali Elkordy [1,*]

[1] School of Pharmacy and Pharmaceutical Sciences, Faculty of Health Sciences and Wellbeing, University of Sunderland, Sunderland SR1 3SD, UK
[2] Department of Pharmaceutics, Faculty of Pharmacy, Prince Sattam Bin Abdulaziz University, Al-Kharj 11942, Saudi Arabia
[3] Department of Pharmaceutics and Industrial Pharmacy, Faculty of Pharmacy, Beni-Suef University, Beni-Suef 62514, Egypt
[*] Correspondence: amal.elkordy@sunderland.ac.uk

**Abstract:** Parkinson's disease (PD) is a progressive neurodegenerative disease that is characterized by the loss of dopamine. Since dopamine has trouble entering the blood–brain barrier, the utilization of dendrimers and other nanomaterials is considered for conjugating the neurotransmitter and other PD drugs. Dendrimers are three-dimensional, hyper-branched structures that are categorized into several generations. Alpha-synuclein (ASN) is the protein involved in regulating dopaminergic functions and is the main aggregate found inside Lewy bodies. Different types of dendrimers have shown efficacy in disrupting the formation of unstable beta structures of ASN and fibrillation. The conjugation of PD drugs into nanomaterials has elicited a prolonged duration of action and sustained release of the drugs inside the BBB. The objectives of this study are to review the applications of a dendrimer-based drug delivery system in addressing the root cause of Parkinson's disease and to emphasize the delivery of anti-Parkinson's drugs such as rotigotine, pramipexole and dopamine using routes of administration other than oral and intravenous.

**Keywords:** Parkinson's disease; neurodegenerative; alpha-synuclein; dendrimer; fibrillation

## 1. Introduction

Parkinson's disease (PD) is a chronic, progressive neurodegenerative disease of the central nervous system that affects the initiation and execution of voluntary movements as well as cognitive impairment [1]. It is considered one of the most common neurodegenerative diseases second to Alzheimer's disease. The main morphological alteration common to all forms of PD is the loss of dopamine due to neuronal degeneration and the loss of melanin-containing nerve cells of the substantia nigra pars compacta (SNpc) [2]. The cardinal symptoms such as tremors, problems with balance and posture, slowing of movement or bradykinesia, and stiffness of peripherals become more prominent for people beyond 60 years [3]. The onset of symptoms is a cascade of events. The dopaminergic pathway starts to degenerate due to a subsequent loss of SNpc neurons. This then results in a substantial decrease in the amount of dopamine that the brain normally produces.

In neurodegenerative diseases, the most difficult concept is the uncertainty of the origin of the process of degeneration [4]. The study of proteins became relevant to neurological problems such as Parkinson's disease because of the findings of protein aggregates in the brain. This aggregation then imposes a need for answers on how it started and what could possibly trigger such events [5]. In fact, all progressive neurodegenerative diseases are originated from the unusual aggregation of proteins, proteotoxic stress or what we know as errors in the protein synthesis and oxidative stress which proteins undergo when they are unstable [6]. Naturally, when a substance is deemed unstable, it tends to share the instability with adjacent pathways involved. This is what makes neurodegenerative

diseases so interesting because they all started with the simple misbehavior of proteins, gradually leading to a damaged pathway, thus the term "degeneration". A case in point is a study by Michel et al. [7] that presented evidence that the SNpc dopamine neurons had degeneration and can have a progressive degradation due to misfolding and aggregation of ASN in the synapse. With regard to the aforementioned alpha synuclein or α-syn, it is a protein thought to regulate the voluntary and involuntary release of dopamine and is vital in understanding how the degeneration started [8].

Dugger and Dickson [6] labeled a disorder termed α-synucleinopathies as a phenomenon where α-synuclein aggregates inside the Lewy bodies. As mentioned in the historical Pearce [9] study, Lewy already found that there is a clump of strange circular matter in the cytoplasm and this inside matter is hereby identified as α-synuclein. Back in 2017, research articles still claimed that there were no specific biomarkers to diagnose PD. However, with the breakthroughs and advancements of today's medicine, the potential of α-synuclein as the official biomarker for PD is now being investigated.

Nanotechnology is a multifaceted aspect of science that covers materials and devices in the nanometer (nm) dimension [10]. It is attracting attention in pharmaceutical research due to the versatility of its applications. Amongst all the substances that nanotechnology covers, the most relevant to this study is the concept of nanomaterials.

This study aims to review the applications of nanoparticles focusing on dendrimer-based delivery systems of large molecules addressing Parkinson's disease. It intends to emphasize the delivery of anti-Parkinson's drugs such as rotigotine, pramipexole, and dopamine. This review also aims to discover how these drug-dendrimer conjugations better help in targeting the aggregates in the brain that cause an imbalance in the homeostatic process and lead to PD. Moreover, this review will attempt to review the advantages and limitations of nanoparticle-based drug delivery systems in addressing PD and to discover other innovative solutions not only to PD but to neurodegeneration in general.

## 2. Blood–Brain Barrier Pathway: An Overview of the Problem

In the human body, blood is able to circulate with the help and delivery of blood vessels. The blood–brain barrier or BBB as its name implies serves as the barrier for every substance that wants to cross borders to the brain from the blood. It is composed of a single layer of polarized endothelial cells (ECs) and mural cells that are classified as continuous and non-fenestrated which is responsible for regulating the central nervous system (CNS) homeostasis [11]. There is also the presence of efflux transport proteins such as p-glycoproteins and multidrug-resistant protein-1 (MRP-1) [12]. The CNS is also protected from unwanted toxins and ions that have the potential to alter neurological activity due to the presence of pores in the nanometer size range [13].

Regrettably, most pharmacologically approved treatment for CNS disorders is not able to perform their activities due to the barrier that protects the brain from unwanted substances. These drugs are usually macromolecules that are either unable to enter the BBB or able to cross but not in a pharmacologically significant amount.

Among the several parts of the BBB, the most significant to this review is the tight junctions (TJs). TJs bind the layers of the endothelial cells together allowing the formation of a highly resistant paracellular barrier and a transendothelial electrical resistance (TEER) to substances crossing the BBB [14]. This limits homeostatic ions such as potassium ($K^+$), calcium (Ca), sodium ($Na^+$), and other molecules to pass through intercellular spaces [12]. However, studies by Daneman and Prat [11] and Van Itallie et al. [15] mentioned specific junction discontinuities which allow selective passage of unionized molecules having a size not greater than 4 nm. Through this evidence, it can be concluded that large molecules have a chance to penetrate the BBB through the presence of these TJs' discontinuities given that support via conjugation of materials with a size not greater than 4 nm. Figure 1 illustrates the general proposed mechanisms of molecules' transport across the BBB.

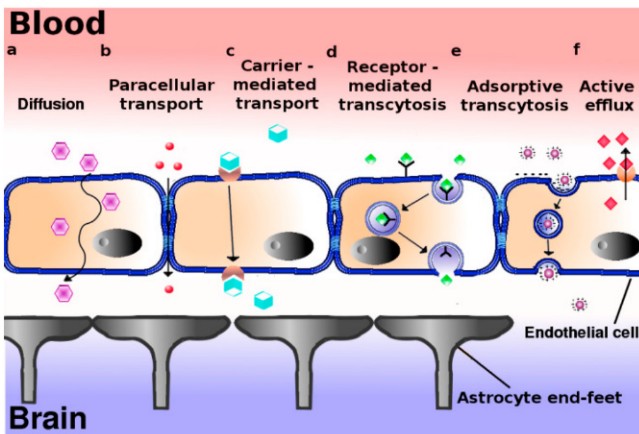

**Figure 1.** Proposed mechanisms of transport across the blood–brain barrier. The blood–brain barrier consists in essence of a polarized layer of vascular endothelial cells, tightly attached to each other by means of tight junctions, and lined up by astrocytes. A variety of transcellular transport processes can be distinguished: (**a**) Diffusion, driven by a concentration gradient, mainly involving small hydrophobic molecules. This pathway represents the main entry route into the brain of current therapeutics; (**b**) Paracellular transport–limited to small water-soluble molecules; (**c**) Carrier-mediated transport, as occurs for, e.g., glucose, amino acids, nucleosides, and therapeutics such as vinca alkaloids, azidothymidine, etc.; (**d**) Receptor-mediated transcytosis for peptidic signaling and regulatory molecules (insulin, leptin, interleukins), nutrients (iron, LDL); (**e**) Adsorptive transcytosis, presumably relying on the transport of positively charged cargo (serum proteins) in a non-specific manner; (**f**) Proton pump efflux transporters. Reprinted from [16], MDPI, 2014.

## 3. The Role of Dopamine and the Dopamine Receptors

Since dopamine is not at its normal level, the natural response of the brain in circumstances such as PD is that it tries to activate the dopamine receptors to produce dopamine like what normally takes place. The conflict with PD is that the SNpc already has a problem since it started deteriorating or degrading, which is why even if the brain instructs it to produce a normal amount of dopamine, it cannot supply the same amount anymore since there is something wrong with its function. This is where anti-Parkinson's drugs come in specifically levodopa and most of the time in combination with other active ingredients to exert the effect [17]. Levodopa, commonly termed L-dopa, is the lipophilic precursor of the BBB-problematic dopamine which remains to be the gold standard of PD symptomatology treatment [18]. In practice, levodopa is given together with Carbidopa [19], a peripheral dopa decarboxylase inhibitor that prevents the premature conversion of levodopa to dopamine because the latter cannot enter the blood–brain barrier [20]. By combining carbidopa with levodopa, the amount of L-dopa being delivered to the brain increases significantly but it is important to take note that carbidopa is not responsible for the increase in dopamine since it mainly serves as support to levodopa and does not perform any synergism [20]. Aside from the levodopa–carbidopa combination, some of the anti-PD drugs under study are rotigotine, selegiline, and pramipexole.

However, in a study by Li et al. [21], they reported that although dopamine is unable to cross the BBB, the input of dopamine in addressing PD is a better option instead of administering levodopa. This is primarily because the receptor responsible for converting levodopa to dopamine can only do so much, especially if the PD case is still in its early phase. Administering dopamine, the primary neurotransmitter lacking in PD, is a direct solution to a progressive problem. Conversely, it is known that dopamine releases reactive oxygen species (ROS) when oxidized that in turn causes further degeneration to an already deteriorating dopaminergic pathway. However, this phenomenon only happens when dopamine is left outside the BBB waiting to be oxidized by monoamine oxidase and not all metabolites of dopamine cause oxidation. To solve this dilemma, there is a possibility of utilizing a nanomaterial to identify and select specifically the beneficial

neuroprotective dopamine metabolite to target the receptors. Nevertheless, dopamine can also be administered by bypassing the BBB by conjugating dopamine to a nanomaterial usually immunoliposome [22], exosomes [23], nanoparticles [24,25] or with regard to this review, dendrimers.

## 4. Dendrimers

According to Rekha and Sharma [26], generally in pharmaceutics, drugs that have difficulties in exerting their pharmacological effects can be incorporated with carriers or conjugated with another drug to synergize the total effect. Especially in cancer therapy, the specificity of targeting is of extreme importance due to the status of the cells. For example, chemotherapeutic agents are better performed with a nano-carrier so that they can target specific cells or sites while bypassing normal cells that can be potential targets without the nanomaterial attached [27]. Therapies currently available for various CNS diseases are given through invasive and non-invasive techniques with the latter deemed as safer and more cost-effective as compared to a high-risk invasive procedure [28]. Amongst non-invasive approaches, the most commonly used nanomaterial in neurodegenerative diseases are nanoparticles (NPs) which are under a bigger umbrella of colloidal drug carriers [29,30]. Recently, De Marco [31] reviewed the application of supercritical fluid technology to produce NPs.

Since PD is the focus of this review, studies are being made to investigate which nanomaterial is the most compatible and most efficient in addressing the symptoms. Some of the nanomaterials are nanoparticles, immunoliposomes, and dendrimers [29,32,33]. Dendrimers are highly branched molecules having a distinct 3D structure with low dispersion yet high performance [34]. Dendrimers are well known to have the capacity of loading high concentrations of drugs and transporting them to biological membranes through endocytosis. In 2002, a substantial amount of research showed several drug studies of conjugation to dendrimer but most of those studies are specific to cancer therapy. Basically, dendrimers can be categorized into generations. These generations evolve over time because of the amine functional core. This review is limited to types of dendrimers that have been used or have the potential to be used for Parkinson's disease.

Firstly, in the case of a polyamidoamine or PAMAM dendrimer, its core undergoes a process called Michael addition reaction with methyl acrylate. After the addition, each amine group gives birth to two ester-terminated branches called half-generation. As the process continues in a cycle of amidation, a full generation will then be achieved, thus having several newer dendrimer generations. This process explains the "generational synthesis" of dendrimers due to a continuous synthesis and production of newer generation after generation [35]. The PAMAM dendrimer is widely used for its antimicrobial, antioxidant and antiviral drug delivery mechanism [36–38].

Secondly, the polypropyleneimine or PPI dendrimer (Figure 2 shows PPI dendrimers decorated with glycerol derivatives) is the most well-known dendrimer commercially. This is usually employed for hydrophobic drugs since its amine terminal components have the capability of increasing the solubility of the enclosed drug/s. However, due to its cationic nature on the surface, it causes lysis of the cells, hence the need for "pegylation" an addition of polyethylene glycol (PEG) [33]. PPI dendrimer's ability to cause cell damage provokes quite a few modifications to correct its disadvantages [39,40].

Lastly, the poly-L-lysine (PLL) dendrimer or dendri-grafted-poly-L-lysine [33] is the better version of both the PAMAM and PPI due to its enhanced biocompatibility, safer drug delivery, avoidance of enzyme degradation, and the additional application in gene delivery.

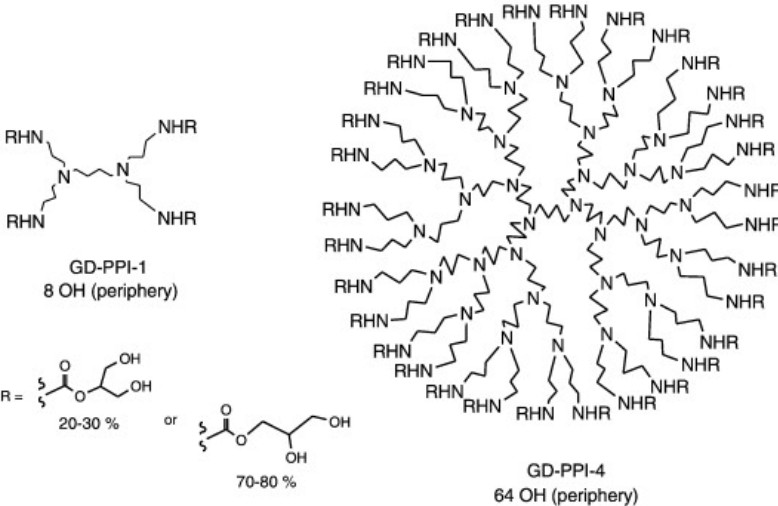

**Figure 2.** Poly(propylene imine) dendrimers decorated with glycerol derivatives: first generation (GD-PPI-1) and fourth generation (GD-PPI-4). Reprinted from [41], Elsevier, 2013.

## 5. Suitability of Dendrimers for Parkinson's Disease

In the study of Milowska et al. [42], it was discussed that ASN is known to help in regulating the dopaminergic system as well as synaptic function. Although its exact physiological function is still unknown, it is apparent that this protein can modify its conformation which can lead to aggregation later on. Its connection to several neurological disorders such as Parkinson's disease is justified by its presence on Lewy bodies as aggregates or clumps. As briefly discussed earlier, the dendrimers in focus in these articles are PAMAM generation 4 or PAMAM G4 dendrimer [42], carbosilane dendrimer [43], and three separate generations of cationic pyridylphenylene dendrimers [44]. Characterization tests were also identified to highlight any similarities between Thioflavin-T fluorescence, circular dichroism spectroscopy, and other microscopic techniques, particularly transmission electron microscopy (TEM). The utilization of TEM helped in understanding the morphology of the prepared dendrimer [45]. Their activity and effectiveness against the fibrillation and capability to aggregate were tested against human ASN, rotenone-treated hippocampal mouse cell line, and ovine particular prion protein inclusion bodies (PrP IB) but these in vitro tests details will not be covered in full synthesis on this study.

In the study regarding carbosilane dendrimers by Milowska et al. [43], a substantial difference between the dendrimer's effect and rotenone was observed with regard to the mitochondrial membrane potential and the amount of reactive oxygen species (ROS) released. Rotenone is a commonly used pesticide that has the capability to inflict damage to the mitochondria complex I. Its lipophilic nature allows it to enter the BBB quickly. This finding is deemed essential in this review as it will help in linking the use of dendrimers as a potential antioxidant. As a matter of fact, its intervention in the PD-linked rotenone-damaged cells recorded an approximately 90% cell viability for the brain-dopamine rotenone cell 7 (BDBR7) and 83% for brain-dopamine rotenone cell 11 (BDBR11) whilst rotenone had only 63%. BDBR7 and BDBR11 are the two types of dendrimers used in the featured study. From these numbers, it can be deduced that rotenone-damaged cells have already produced a substantial amount of ROS in the brain which causes the mitochondria to fail due to imminent cell death by oxidation. To establish the relationship clearly, the higher the ROS present in the cells, the lower the cell viability will be. Furthermore, the more cells exposed to oxidation, the higher the chances of cell death. This in turn will then make the disease a lot worse because there will be none left to metabolize and process whatever anti-Parkinson's drug will be used.

The mentioned articles so far were consistent in continuously mentioning incubation and pre-incubation of the prepared dendrimers while doing their characterization tests. The reason for the incubation of dendrimers is for assessing the thermodynamic stability of

the prepared formulation. On the other hand, the pre-incubation of dendrimers together with ASN is to monitor whether dendrimers can still affect a heat-denatured ASN.

## 6. Anti-Parkinson's Drugs and Dopamine

### 6.1. Rotigotine

In the study of Choudhury et al. [46], the primary objective is to formulate a stable nanoemulsion version of rotigotine, a non-ergot dopamine agonist which is famously marketed as a Neupro skin patch [47]. The main goal is to successfully prepare an adhesive to be applied to a mucosal surface of the body as a manner of drug delivery. In this study, the technique performed in preparing the rotigotine mucoadhesive nanoemulsion is via an aqueous titration method where the active drug, rotigotine, is solubilized in Capryol 90. Supposedly, a rule in formulating nanoemulsions is that the selected solubilizing agent should be the material where the API is most soluble. In this case, the solubilizing agent is Transcutol HP. However, this study found out that although rotigotine is most soluble in Transcutol HP, it will not yield a stable nanoemulsion for the long run, hence the selection of Capryol 90 as an alternative. Moreover, the selection of a suitable rotigotine-loaded nanoemulsion (RNE) to be prepared as a mucoadhesive nanoemulsion is based on the droplet size, polydispersity index (PDI), and most importantly thermodynamic assessment. After subjecting the chosen rotigotine nanoemulsion (RNE) to various stress tests, it was found that the prepared rotigotine mucoadhesive nanoemulsion final preparation (RMNEF) which is derived from the RNE is capable of releasing the API slowly while yielding a very high mucoadhesive strength.

### 6.2. Pramipexole

In the study by Raj et al. [45], the role of chitosan in a nano-sized anti-Parkinson's drug has been highlighted immensely. The technique of incorporating chitosan with pramipexole is via the ionic gelation method wherein the cation amino group in chitosan interacts with the presence of the negatively charged components of sodium tripolyphosphate (STPP). Chitosan imparted a cationic charge to the formulated pramipexole nanoparticle (P-CN). Since the cell membrane that it attaches to is negatively charged (anion), this gives the advantages of prolonged duration inside the body and improved absorbance. After the morphological evaluation using the transmission electron microscope (TEM), it was revealed that P-CN had a spherical shape which is also beneficial for its improved flow along the bloodstream and during its perfusion to the BBB. According to Okura et al. [48], the uptake of pramipexole into the BBB is via a so-called organic cation-sensitive transporter. Since chitosan has a cationic charge and the manner by which pramipexole is being transported to the BBB is via a cationic transporter, this only proves the advantage that chitosan imparted to the formulation because of the similarity of their charges. In this study, the antioxidant activity of pramipexole was also discussed via the measurement of reduced glutathione (GSH) levels. GSH is one of the major antioxidants naturally found in the body and plays a major role in terms of analyzing neurodegenerative diseases such as PD [49]. Raj et al. [45] finally reported that the administration of P-CN on rotenone-damaged cells has prevented the decline of GSH in the brain cells of the rats via a postmortem analysis. Interestingly, it was previously established in this study that intranasal administration of the formulated P-CN yielded the most improved increase in dopamine levels. However, as far as the antioxidant effect of pramipexole is concerned, the oral version of pramipexole caused the highest improvement in the GSH levels.

### 6.3. Dopamine

Two research articles specific to dopamine were included in this review to emphasize the assistance of a nanoparticle on the crossing of dopamine to the BBB whilst avoiding oxidation in the periphery. Kang et al. [32] have utilized immunoliposomes and polyethylene glycol-assisted (PEGylated) immunoliposome. To demonstrate whether dopamine has been successfully delivered through the BBB, pharmacokinetics testing needs to be

performed because clinically speaking, the most optimal way to measure whether the drug is working or not is to observe the very same parameters that can declare its effect. As shown in Figure 3 below, it was found that the conjugation of dopamine with PEGylated immunoliposome had managed to reduce the clearance of dopamine to only a range of 0.15 to 0.21 mL/min/kg as compared to the free dopamine's clearance of 1.98 to 2.64 mL/min/kg [32], meaning that Figure 3 is the plot of the plasma concentration against time profile of the three formulations featured in the study [32] where the free DA showed a rapid decrease in the concentration and maintained decreasing over time. On the other hand, both DA-PL and DA-PIL conjugates had retained a high concentration over the given time period which indicates a now modified low clearance for the DA when assisted with PEGylated immunoliposome. This pharmacokinetics data is supported by the measurement of the area under the curve (AUC) of a similar study which reported that Dopamine-PEGylated immunoliposome (DA-PIL) yielded 516 to 890 min/mL AUC as compared to only 39 to 51 min/mL AUC for the free dopamine. What interests our review are the findings on the relationship of incubation time to the physical stability of the formulated DA-PIL, after 24 h of incubation, 67% of dopamine managed to remain; however, the amount of dopamine gradually decreased over the period of 72 h with 51% left. Although the numbers are not very far from each other, we can hypothesize that prolonged exposure to heat definitely changed the conformation of the formed protein combination and perhaps even had an interaction with the rat plasma to which it was exposed.

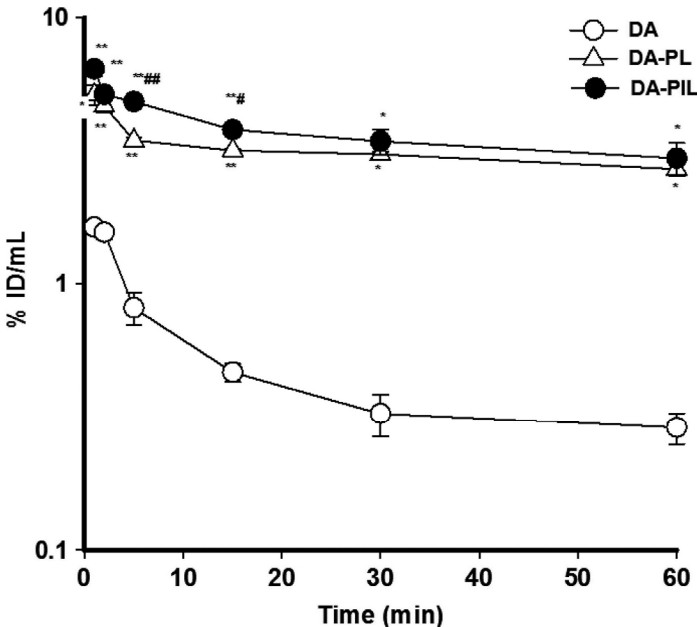

**Figure 3.** The percentage of injected dose per milliliter of the plasma–time profiles of DA (○), encapsulated DA-PL (△), and DA-PIL (●) in PD rats up to 60 min after intravenous injection. DA, free dopamine; DA-PLs, encapsulated dopamine PEGylated liposomes; DA-PILs, encapsulated dopamine PEGylated immunoliposomes. Data are means ± SEM (*n* = 3 or 4). * $p < 0.01$; ** $p < 0.001$ significantly different from DA. # $p < 0.05$, ## $p < 0.01$, significantly different from DA-PL. Reprinted from [32], Wiley, 2016.

## 7. Discussion

It has been long established that the delivery of drugs to the central nervous system does not always end at ease due to the very protective blood–brain barrier. The BBB follows certain criteria based on the physicochemical characteristics of the drug. Since almost all of the drugs purposely made for targeting various neurological diseases have issues with the BBB, the nano-sizing and conjugation of these drugs to several nanomaterials are now of

high interest. In this review, the usage of dendrimers as nano-carriers of anti-Parkinson's drugs and generations of dendrimers are discussed.

Dendrimers can assist the passage of both small and large-molecule drugs into the BBB. This is due to its unique hyperbranched structure and its ability to be perfectly modified to suit the nature of the drug [33]. This characteristic of the dendrimer makes it very suitable for PD study because it is proven to inhibit the fibrillation of ASN [50]. In this way, when the instability itself is addressed, the disease will no longer be considered progressive because the dendrimers already dealt with the protein's instability. In a study by Mignani et al. [51], dendrimers are not only used as nano-carriers but also as anti-amyloidogenic, anti-prion agents, and an inhibitor of alpha-synuclein. Its application is not only limited to cancer therapy but also to several neurodegenerative diseases [52] and disorders which are the interests of this study.

### 7.1. Characterization Tests and Materials

By looking at the materials and methods summarized in Table 1, we can observe that there is a consistency in the inclusion of dimethylsulfoxide or better known as DMSO. Although the specific function of DMSO in those studies is not specified, it is safe to hypothesize that it might have been used to assist in disrupting the BBB via biochemical disruption. According to Mignani and Pierre-Majoral [53], since the BBB is a rate-limiting step in the pathway of most neuronal drugs, there are two common ways to disrupt it. This includes biochemical disruption and the less recommended osmotic disruption. DMSO happens to be classified as a biochemical agent for disruption due to its ability to reduce the integrity of the endothelial cells that can be found in the BBB therefore increasing the permeability of drugs. This rationale is backed up by research conducted by Kleindienst et al. [54] where they found that even a low concentration of DMSO (1 mg/kg) was capable of opening the ischemic side of the BBB, thus resulting in an increased concentration of drug diffusing to the BBB.

**Table 1.** Summary of the characteristics of the 8 selected research articles to review in this study.

| Research Articles | Materials and Methods | Results |
|---|---|---|
| PAMAM G4 dendrimers affect the aggregation of α-synuclein | <ul><li>Human ASN, phosphate buffer saline solution, polyamidoamine (PAMAM) G4 and PAMAM G3.5 dendrimers.</li><li>Intrinsic tyrosine fluorescence, Circular Dichroism (CD) spectroscopy, and Thioflavin-T fluorescence measurement.</li></ul> | <ul><li>PAMAM G4 has increased the fluorescence intensity of the ASN during the Tyrosine fluorescence test and there were no changes incurred by the PAMAM G3.5. PAMAM G4's increase in intensity is probably an interaction between the amino group of dendrimers and the hydroxyl group of tyrosine.</li><li>PAMAM G4 altered the shape of the ASN during the CD spectroscopy after 48 h of incubation.</li><li>PAMAM G4 has shown evidence of its activity against protein aggregation and fibrillation [42].</li></ul> |

| Research Articles | Materials and Methods | Results |
|---|---|---|
| Carbosilane dendrimers inhibit α-synuclein fibrillation and prevent cells from rotenone-induced damage | • Carbosilane dendrimer, Rotenone, dimethylsulfoxide (DMSO), phosphate buffer saline tablets, fetal bovine serum, and trypsin.<br>• The study labeled the dendrimers as BDBR7 and BDBR11.<br>• Measurement of zeta potential, Thioflavin-T fluorescence measurement, CD spectroscopy, measurement of ROS, and mitochondrial membrane potential assessment. | • Carbosilane dendrimers inhibited alpha synuclein fibrillation by up to 91.8–96.7% [43].<br>• Because both dendrimers achieved positive charges, it was revealed that it successfully inhibited fibril formation as it did not progress into the formation of beta structures as compared to its ASN progression counterpart.<br>• Rotenone obviously increased the production of ROS, thus decreasing the membrane potential of the mitochondria, hence cell death. On the other hand, carbosilane dendrimers managed to increase the membrane potential. However, it decreased cell viability but when pre-incubated versions of dendrimer are used, it reverses the issue with the viability of the cells. |
| Disruption of Amyloid Prion Protein Aggregates by Cationic Pyridylphenylene Dendrimers | • Mouse monoclonal anti-PrP antibodies 66.100b3 specific for sequence K26RPKP30 of PrP N-terminus, salts, and buffers.<br>• Synthesis of cationic dendrimers, dynamic light scattering measurement using zetasizer, Thioflavin-T fluorescence, western blot, and fluorescence microscopy. | • The incorporation of dendrimers showed no positive interaction of dendrimers to antibodies, therefore not showing any aggregation. On the other hand, Dot Blot analysis showed that anti-PrP antibodies had an interaction with the IBs [44]. |
| Trans-Blood–Brain Barrier Delivery of Dopamine-Loaded Nanoparticles Reverses Functional Deficits in Parkinsonian Rats | • Poly (D,L-lactide-co-glycolide) lactide:glycolide or PLGA with equal 50:50 ratio, dopamine hydrochloride, homovanillic acid, dimethyl sulfoxide (DMSO), and other chemicals for in vitro testing *.<br>• Preparation of dopamine nanoparticles was via the double emulsion solvent evaporation method.<br>• Characterization tests involved are size analysis via zeta potential, microscopic examination through TEM and scanning electron microscope (SEM), and determination of drug entrapment efficacy and drug computation of drug loading. | • The particle size of the prepared dopamine NPs is smaller when measured via TEM and bigger when measured in dynamic light scattering (DLS).<br>• Bulk dopamine is more prone to oxidation and produces more ROS as compared to the nanoparticle form. The bulk dopamine also resulted in the highest number of cell death from several test concentrations as compared to both PLGA NPs without dopamine and PLGA NPs with dopamine [25]. |

**Table 1.** *Cont.*

| Research Articles | Materials and Methods | Results |
|---|---|---|
| Formulation development and evaluation of rotigotine mucoadhesive nanoemulsion for intranasal delivery | • Rotigotine, Capryol 90, ethanol, Tween 20 and other excipients.<br>• Preparation of mucoadhesive nanoemulsion via titration method.<br>• Characterization tests involved are TEM, zeta potential measurement, viscosity determination and mucoadhesive strength test among others.<br>• Tested on the nasal mucosa of goat *. | • RNE1 (origin of RMNEF) yielded the lowest droplet size of only 44 +/− 2 nm while the final RMNEF preparation yielded an even lower size of 130 +/− 24 nm and a positively charged zeta potential.<br>• RNE1 released a large amount of rotigotine at 89.29% within 8 h while RMNEF slowly released 70.73% within 8 h [46].<br>• The prepared RMNEF was concluded to have Korsmeyer–Peppas release kinetics. |
| Pramipexole dihydrochloride loaded chitosan nanoparticles for nose to brain delivery: Development, characterization and in vivo anti-Parkinson's activity | • Pramipexole dihydrochloride, dopamine, chitosan, sodium tripolyphosphate<br>• Formulation of Pramipexole-Chitosan nanoparticle (P-CN) via ionic gelation method.<br>• Characterization tests involved are the measurement of zeta potential and size using DLS, percentage drug efficacy and percent drug entrapment efficiency measurement, and microscopic examination (TEM, SEM), and several in vitro assays.<br>• Tested on male Sprague–Dawley rats *. | • Physically, there was an increase in size when the API was added and upon TEM's observation, there was roughness observed on the surface of the P-CN.<br>• After in vivo testing, dopamine levels of the rat group with rotenone (aka disease group) had only 48.33 +/−3.57 ng/g tissue; the treated with Pramipexole intranasal solution had 81.61 +/−4.44 ng/g tissue while the group treated with the prepared P-CN had 97.38 +/−3.91 ng/g tissue [45].<br>• The study also investigated the antioxidant activities of pramipexole. |
| Use of PEGylated Immunoliposomes to Deliver Dopamine across the Blood–Brain Barrier in a Rat Model of Parkinson's disease | • Dopamine, Distearoylphosphatidylcholine, monoclonal antibodies against OX26, cholesterol, and other excipients *.<br>• Preparation of PEGylated liposome and immunoliposome via evaporation and extrusion.<br>• Characterization methods involved physical stability testing, internal carotid artery perfusion (ICAP) method (as in vivo *), and brain distribution assessment for dopamine. | • The area under the curve (AUC) of the PEGylated liposome and PEGylated immunoliposome increased by 14 and 16 times higher, respectively, than the AUC for the free dopamine [32].<br>• The duration of stay of the PEGylated liposome and immunoliposome have also increased with 116 and 107 min long stay as compared to the 45.6-min half-life of free dopamine. |

Table 1. *Cont.*

| Research Articles | Materials and Methods | Results |
| --- | --- | --- |
| Pharmacokinetics and pharmacodynamics of intranasally administered selegiline nanoparticles with improved brain delivery in Parkinson's disease | • Selegiline, rotenone, chitosan (CHS), and sodium tripolyphosphate (STPP).<br>• Formulation of the intranasal selegiline nanoparticles using ionic gelation method.<br>• Characterization tests involved are size analysis via zeta potential assay, microscopic examination via TEM, and calculation of the drug entrapment efficacy. In vitro testing was also conducted.<br>• Tested both on the nasal mucosa of goat and on male Sprague–Dawley rats *. | • The combination of CHS and STPP has the smallest droplet size, lowest polydispersity index (PDI), and the highest amount of drug entrapped in its matrix [55].<br>• Evidently, the preparation with the highest amount detected on both brain and plasma concentration is the nanoparticle version of selegiline via the intranasal route (3.93 ng/g and 4.27 ng/g, respectively.)<br>• The maximum concentration or $C_{max}$ of selegiline NPs increased by 12 times higher than the conventional oral dosage form and route of administration. |

* Refer to corresponding articles for other chemicals which are purposely excluded in this table since the tests which they are used are not the focus and are considered beyond the coverage of this review. In the results presented in the previous chapter, three specific articles utilized dendrimers in their research. In the research of Milowska et al. [43], cationic carbosilane dendrimers were featured and showed promising effects against the fibrillation of alpha-synuclein. This fibrillation process happens when the ASN starts to become unstable and changes its conformation into a beta harmonica structure, thus causing aggregation and clusters. In another stduy by Milowska et al. [42], a comparison between PAMAM 3.5 and 4th generation was performed with emphasis on their impact on ASN aggregation while observing the dendrimers' activity on circular dichroism (CD) spectra and relating it to their effect against ASN. Lastly, Sorokina et al. [44] highlighted the significance of pre-incubation of the inclusion bodies or the proteins to the prepared dendrimers to allow a premature degradation of the aggregates due to their exposure to the dendrimers. In this way, the effects of the dendrimers on the aggregates is more visible.

Another concept in the featured literature that interests this study is the CD spectroscopy results of the dendrimers. CD spectroscopy is performed to identify any conformational changes in the secondary structure of proteins [56]. In the study, CD spectroscopy was used to check whether there were changes in the ASN's conformation and behavior when subjected to the PAMAM dendrimers. The rationale for why CD spectroscopy was used is because of the rapid turnover of the results especially when the sample to be tested contains 20 micrograms or less of the sample, which in the study only used 2 micromoles. Along with their findings, Milowska et al. [42] stated that there was no positive signal at 195 to 206 nm wavelength for the ASN with PAMAM G4 dendrimers while the ASN alone had shown a positive signal on the same wavelength range. The positive signal on the ASN spectra alone indicates the formation of the beta structure. Consequently, the beta structure has negative results at 218 nm and positive at 195 nm [56]. The presence of a positive signal on the ASN spectra alone only means that after 48 h of incubation, ASN already manifested aggregation because its structure has changed its conformation, hence forming the beta harmonica structure which further leads to the formation of soluble oligomers, thus starting fibrillation. To establish the relationship clearly, when a protein is exposed to heat, which in this case is the incubation period, it starts to alter its conformation as it denatures, and while in that process, the natural behavior of the protein goes unstable which leads to problems, which again, in this case, is degeneration of the surrounding cells in the dopaminergic pathway that obviates dopamine supply. Because of this, the absence then of a positive signal on the incorporation of PAMAM G4 to the ASN only confirms that G4 dendrimer has inhibited fibril formation. However, CD spectroscopy is not the only method of identifying secondary structures. There is also Raman spectroscopy and Fourier Transform Infrared (FTIR).

Lastly, it is also important to note the significance of the Thioflavin T fluorescence in the study. In Figure 4 below, the sole ASN shows high fluorescence intensity while two

concentrations of ASN + PAMAM G4 have lower intensity. This reflects the idea that their intensity depends on the presence of aggregates. However, in this study, it was stated in Table 1 that PAMAM G4 dendrimers increased the fluorescence intensity of the ASN [42]. Observing from the graph below, it is evident though that the ASN alone only showed the highest intensity as it progresses proportionally with time. From this, it is safe to hypothesize that the longer the ASN is subjected to heat (increasing time on the x-axis), the more aggregation it creates. Nevertheless, both concentrations of the dendrimer showed activity against ASN but notice that the higher concentration only exhibited most of its activity against ASN after incubation at around the 40th to 50th hour and became similar to the lower concentration at the 72nd hour onwards. To establish the rationale, the longer both proteins are subjected to heat, the more unstable they become. The only difference is that ASN alone is creating more aggregates as its exposure is prolonged, hence increasing its intensity, while the dendrimers elicit "structural reorganization" [42], that is inhibition of aggregation, at the early phase then becomes steady after quite some time. It can also be deducted from this graph that the PAMAM G4 at the lower concentration had a more rapid interaction with the ASN as compared to the higher concentration.

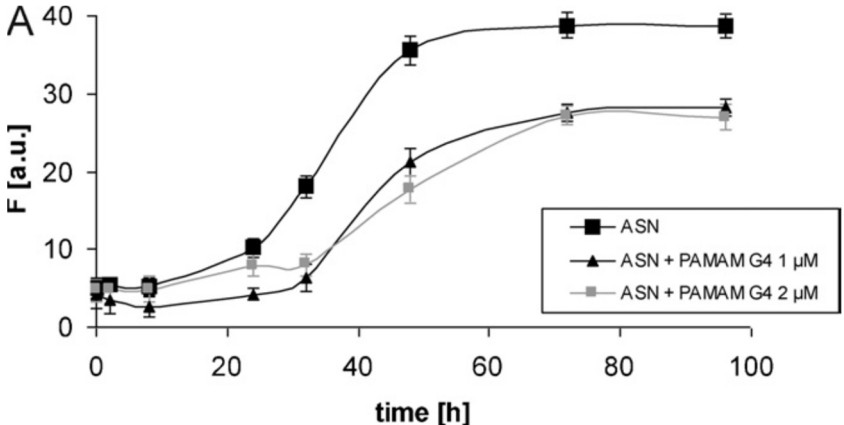

**Figure 4.** Plot of the Thioflavin T fluorescence assay result showing different concentrations of PAMAM G4 dendrimer derived from [42], Elsevier, 2011.

*7.2. Korsmeyer-Peppas Reaction Kinetics*

In any drug studies, the kinetics of the drug is a vital part of experimentation to analyze how fast or slow it releases in its target site in the body or whether it is present in the plasma which is another important parameter for further studies. In terms of one of the drugs featured, rotigotine, the researchers described its release as having a mathematical model named Korsmeyer–Peppas release kinetics [46]. Some of the more common mathematical models used in the pharmaceutical industry that we know are Higuchi, first-order, and zero-order. The Korsmeyer–Peppas release kinetics (equation 1 below) is used in identifying the kinetics of drugs having modified release. The $C_t/C_\infty$ represents the ratio of the drug release over a unit of time while $k$ is the rate constant, $t$ is the time and $n$ is the transport exponent [57].

$$\frac{C_t}{C_\infty} = k \times t^n$$

According to Wu et al. [57], Korsmeyer–Peppas is applicable for drugs that have a non-linear diffusion profile. This profile can be observed for drugs being encapsulated from a matrix, a polymer, and even liposomes [58]. The usage of Korsmeyer–Peppas in the rotigotine study and even in other studies featured in this review is fitting because obviously the drugs are conjugated with nanomaterials. This only means they are encapsulated and therefore will exhibit a controlled release inside the matrix towards the external environment. This motion of releasing indicates the presence of time. Additionally, since

dopamine has an innately rapid half-life, its conjugation to a nanomaterial is very important for prolonging its stay and stability inside the dopaminergic pathway.

### 7.3. Intranasal Route and Chitosan

Since the beginning, there is a continuous search for alternative routes of administration for drugs that have special considerations in terms of their delivery, dosage form, and target site. The intranasal route received massive interest in the research field due to its advantages.

Firstly, the intranasal route has the advantage of delivering the drug better than the oral route because there is no first-pass metabolism for the latter. This is extremely beneficial for drugs that have a very low half-life and sensitive physicochemical features that need avoidance from stomach acids. Another advantage is that it allows the incorporation of macromolecules into mucoadhesive and other bio-adhesive formats because these drug-delivery systems swell or expand when exposed to the nasal mucosa. A great example of the application of these advantages is the delivery of PD drugs. It is inherent in these drugs' nature that they are convenient to be taken orally; however, prolonged oral administration of these drugs brings a lot of disadvantages such as premature peripheral conversion of dopamine thus leading to the manifestation of motor symptoms such as tremors.

In the study of pramipexole, based on its TEM and SEM analysis results, we can hypothesize that the delivery and effect of pramipexole + chitosan via the intranasal route were improved because of the cationic charge that chitosan imparted to the formulation. To reiterate the rationale, the TEM and SEM results revealed the spherical shape of the drug-nanomaterial conjugation which allowed an improved flow inside the BBB because it mimics the shape of most biological components in the nano range such as blood cells. The incorporation of chitosan also improved the characteristics because it increased the permeation of pramipexole and even rotigotine. This then helped in reaching a steady-state flux because of the higher permeability coefficient [46]. To establish its relationship, the positive charge of chitosan gave contrast to the anionic charge of the cell membrane. With its attracting opposite charges, the cell membrane opened the channels, hence allowing more of the drug to diffuse inside and not to be metabolized externally. This also applies to dopamine which has problems with metabolism when left externally. That is why the modification of their route of administration is a necessity to elicit effects inside the cell systematically. The addition of chitosan can be backed up by another concept regarding the polydispersity index (PDI) results of these drugs. Apparently, the lower the PDI of a formulation, the more uniform its droplet size. With the incorporation of chitosan, the overall viscosity of the formulation increased. This elevation of viscosity somehow allowed the formulation to have a prolonged residence time in its target site, thus reducing the amount of eliminated drug concentration in a small span of time.

In the study of Pahuja et al. [25], PLGA, a biodegradable and biocompatible vehicle for drug delivery was used to conjugate dopamine for direct BBB delivery. Fortunately, PLGA provided constant and sustained release while dopamine is entrapped inside. Since there is already degeneration in PD, there is a substantial decrease and absence of available receptors that can transport dopamine from the periphery towards the synapse. This is why matrix-entrapped dopamine had enhanced retention time instead of the usual short half-life of free bulk dopamine. This study also promoted the advantage of PLGA-Dopamine tandem because the entrapment alone meant that dopamine is not available for oxidation; hence, dopamine metabolites will not produce any ROS. The pH of the environment is also another factor for consideration because dopamine favors low pH for its stability.

## 8. Conclusions

Although not all of the featured articles in this study utilized dendrimers as the nano-carriers for the anti-Parkinson's drugs featured, the researcher sees it as an opportunity to explore how those drugs behave in a nano environment. These pieces of literature are specifically included since they have the potential to be conjugated into a dendrimer. Because there are several research studies that conjugate them with other nanomaterials

already, the characterization tests and findings are already established. This will then provide a more stable backbone for future dendrimer conjugation studies.

The articles regarding dopamine administration using a different approach in the BBB are really two innovative approaches since it is the primary neurotransmitter that is essential with PD. These collated data are aimed at presenting how useful and innovative the conjugation of PD drugs to nanomaterials are particularly on the massive impact of dendrimers in preventing the formation of the aggregates in the first place. Personally, the researcher sees it as an outlet that the dendrimers can indeed serve a purpose in terms of addressing the root cause of the aggregation rather than addressing it after the degeneration started. With the gathered data on the dendrimers, we can conclude that different generations of dendrimers have a common aim in the formation of unstable beta structures of ASN. They also have a commonality in the fibrillation process itself like the way these dendrimers interact with ASN based on the spectra findings and how brilliant they are on inhibition.

### 9. Limitations and Recommendations

As much as this study wanted to cover all the important details, there were still parts of the featured research articles and topics which were not covered because they are beyond the objectives and expertise of this study. There is no comparative analysis performed between studies. Some of the featured articles have focused on the formulation of the dendrimer and the drug-nanomaterial conjugate while some thoroughly discussed in vitro results such as pharmacokinetic charts, plasma levels, and post-mortem analysis of experimental animals. This study managed to collate pieces of information as far as protein and pharmaceutical studies are concerned but still, some topics might not be as elaborate as others.

With all the data gathered, this study recommends the application of the theories regarding a dendrimer and drug conjugate into in vitro experimental research to collect a newer set of data in the hope of contributing to the search for the optimal therapy for Parkinson's disease and even to other progressive neurodegenerative diseases present.

**Author Contributions:** Conceptualization, M.B.O. and A.A.E.; methodology, M.B.O.; validation, M.B.O., R.M.Z. and A.A.E.; formal analysis, M.B.O., R.M.Z. and A.A.E.; investigation, M.B.O., R.M.Z. and A.A.E.; resources, M.B.O. and A.A.E.; data curation, M.B.O. and A.A.E.; writing—original draft preparation, M.B.O.; writing—review and editing, R.M.Z. and A.A.E.; visualization, A.A.E.; supervision, A.A.E.; project administration, M.B.O. and A.A.E. All authors have read and agreed to the published version of the manuscript.

**Funding:** This research received no external funding.

**Institutional Review Board Statement:** Not applicable.

**Informed Consent Statement:** Not applicable.

**Conflicts of Interest:** The authors declare no conflict of interest.

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
