# Peer review of "Dendrimers-Based Drug Delivery System: A Novel Approach in Addressing Parkinson’s Disease"

_futurepharmacol, doi:10.3390/futurepharmacol2040027_

Round 1
Reviewer 1 Report
The current version of the manuscript has been improved and. However, there are a few more corrections needed to improve the manuscript.
1. Many spacing, and punctuation marks problem are found in the manuscript.
2. Authors are suggested to use the full form when used for the first time throughout the manuscript.
3. Novelty of the work should be added by the author in the introduction section.
4. Spacing, punctuation marks, grammar, and spelling errors should be reviewed thoroughly. I found so many typos throughout the manuscript.
5. Some references are very old. Please try to include the last 10 years' references. Also, cite these references and these are very relevant for this article. Please include them properly.
Rahman MM, Ferdous KS, Ahmed M. Emerging promise of nanoparticle-based treatment for Parkinson’s disease. Biointerface Res. Appl. Chem. 2020 May 20;10:7135-51.
Rauf A, Rahman M. Potential Therapeutics Against Neurological Disorders: Natural Products-Based Drugs. Frontiers in Pharmacology. 2022:3178.
Rahman MM, Islam MR, Emran TB. Clinically important natural products for Alzheimer's disease. International journal of surgery (London, England). 2022 Jul 31:106807.
Author Response
Many thanks for the valuable comments:
Here is our response:
- Many spacing, and punctuation marks problem are found in the manuscript. All corrected and the manuscript has been proofread.
2. Authors are suggested to use the full form when used for the first time throughout the manuscript. Done throughout the manuscript.
3. Novelty of the work should be added by the author in the introduction section. Please refer to the last paragraph of section 1.
4. Spacing, punctuation marks, grammar, and spelling errors should be reviewed thoroughly. I found so many typos throughout the manuscript. Done via the entire text.
5. Some references are very old. Please try to include the last 10 years' references. Also, cite these references and these are very relevant for this article. Please include them properly. There are recent references added, please refer to section 3. Also, this reference "Rahman MM, Ferdous KS, Ahmed M. Emerging promise of nanoparticle-based treatment for Parkinson’s disease. Biointerface Res. Appl. Chem. 2020 May 20;10:7135-51" has been cited into the text. The other two suggested references are mainly on natural products which are not the focus of the review. The newly added references were cited both in the text and at the end - in the reference list.
Reviewer 2 Report
The review article, entitled as, “Dendrimer Based Drug Delivery System: A Novel Approach in Addressing Parkinson’s Disease” is an interesting document. The topic is interesting, but the data added in the manuscript is not enough to de considered as a comprehensive review of literature.
The segments of a review articles do not include the separate discussion section, until or unless, recommended by the journal.
In neurodegenerative disorders, blood brain barrier has its own importance, hence; it should be discussed in more details. Furthermore, the data related to PD should also be elaborated more comprehensively.
Similarly, there should be more details about different types of dendrimers and their mechanism of drug release, because it is important aspect in the drug delivery, specifically to brain.
Has the authors got permission from the parent journal or authors to add the figure 2 & 3 in the manuscript?
Conclusion should be more precise
Authors may add acknowledgement section.
There are some sentences, which are difficult to understand, so authors should thoroughly revised the manuscript to get rid of such type of text, such as;
“However when it concerns one of the most delicate organs in the body – the brain and the central nervous system (CNS), these blood vessels are specialized”. It is difficult to understand the meaning of this sentence, please re-write it in a meaningful way.
The manuscript should be evaluated for proper use of punctuation
Author Response
Many thanks for the review and here is the response for the comments:
The segments of a review articles do not include the separate discussion section, until or unless, recommended by the journal. The discussion has sections for clarity.
In neurodegenerative disorders, blood brain barrier has its own importance, hence; it should be discussed in more details. Furthermore, the data related to PD should also be elaborated more comprehensively. Done, please refer to section 3, an added section.
Similarly, there should be more details about different types of dendrimers and their mechanism of drug release, because it is important aspect in the drug delivery, specifically to brain. Please refer to section 4.
Has the authors got permission from the parent journal or authors to add the figure 2 & 3 in the manuscript? Yes
Conclusion should be more precise. Done
Authors may add acknowledgement section. Not applicable for this review
There are some sentences, which are difficult to understand, so authors should thoroughly revised the manuscript to get rid of such type of text, such as;
“However when it concerns one of the most delicate organs in the body – the brain and the central nervous system (CNS), these blood vessels are specialized”. It is difficult to understand the meaning of this sentence, please re-write it in a meaningful way. The sentences have been rewritten
The manuscript should be evaluated for proper use of punctuation. The manuscript has been proofread.
Reviewer 3 Report
In this review, the authors studied drug delivery systems based on the use of dendrimers. The paper is interesting and can be published following some revisions.
The references in the text can be indicated as first author et al. avoiding the list of all the authors.
In many cases, nanoparticles that can be used as drug delivery systems have been processed through innovative techniques based on the use of supercritical fluids. See, for example, this recent review on nanoparticles and cancer (https://doi.org/10.3390/mi13091449).
The reference list is constituted of 50 papers. It seems to me a really small number of studied articles considering that the present paper is a review. Please, try to consider a wider range of papers.
Author Response
Many thanks for the review and here is the response:
The references in the text can be indicated as first author et al. avoiding the list of all the authors. Done
In many cases, nanoparticles that can be used as drug delivery systems have been processed through innovative techniques based on the use of supercritical fluids. See, for example, this recent review on nanoparticles and cancer (https://doi.org/10.3390/mi13091449). Added, please refer to section 4, first paragraph.
The reference list is constituted of 50 papers. It seems to me a really small number of studied articles considering that the present paper is a review. Please, try to consider a wider range of papers. Recent relevant articles were added please refer for example to the new added section 3.
Reviewer 4 Report
The review article by Ordonio et al presents a detailed analysis of how nanoparticle-based drug delivery can overcome the blood-brain barrier and allow drugs to reach their targets in the brain.
The article is informative, detailed and easy to read and understand.
Specific comments to be answered may include:
1. Summary figure showing how other drug delivery methods are unable to overcome the blood-brain barrier.
2. Singular graphs of Figures 2 and 3 can be put together and figure numbers can be saved for relevant drawings visually depicting the nanoparticle mechanism.
3. Figure 1 should be more detailed and clean. My recommendation is to draw the structures and not directly copy-paste another model without cleaning up the edges and adjusting size.
4. References to papers by other authors should be cleaned up with Author 1 et al (year), all the authors do not need to be listed as that breaks reading flow.
5. Limitations of the study (section 7) are arbitrary and don't point out the weaknesses. Please elaborate and point out that no comparative analysis between studies was performed.
6. The discussion sections should be more focussed and must not be a repetition of the afore discussed material.
Overall, I believe this review has potential for the PD community and highlights nanoparticle drug delivery. English language minor editing is preferred. I strongly recommend diagrammatic representations to improve citations of this article.
Author Response
Many thanks for the review and here is the response:
1. Summary figure showing how other drug delivery methods are unable to overcome the blood-brain barrier. Please refer to the newly added Figure 1.
2. Singular graphs of Figures 2 and 3 can be put together and figure numbers can be saved for relevant drawings visually depicting the nanoparticle mechanism. From the authors point of view, the inclusion of both figures separately is better for manuscript readability, especially since they were taken from articles after obtaining permission.
3. Figure 1 should be more detailed and clean. My recommendation is to draw the structures and not directly copy-paste another model without cleaning up the edges and adjusting size. Changed to include only one piece.
4. References to papers by other authors should be cleaned up with Author 1 et al (year), all the authors do not need to be listed as that breaks reading flow. Done throughout the text.
5. Limitations of the study (section 7) are arbitrary and don't point out the weaknesses. Please elaborate and point out that no comparative analysis between studies was performed. Done
6. The discussion sections should be more focussed and must not be a repetition of the afore discussed material. Done to our best.
Overall, I believe this review has potential for the PD community and highlights nanoparticle drug delivery. English language minor editing is preferred. I strongly recommend diagrammatic representations to improve citations of this article. The manuscript has been proofread, there are three figures within the manuscript.
Round 2
Reviewer 2 Report
The Manuscript can be accepted in its current form.
Reviewer 3 Report
The paper has been improved and can be accepted in the current form